# To strike or not to strike? an investigation of the determinants of strike participation at the Fridays for Future climate strikes in Switzerland

Viktoria Cologna[1]*, Gea Hoogendoorn[1], Cameron Brick[2]

1 Institute for Environmental Decisions, ETH Zurich, Zurich, Switzerland, 2 Department of Psychology, University of Amsterdam, Amsterdam, Netherlands

* viktoria.cologna@hest.ethz.ch

## Abstract

The Fridays for Future strikes involve students striking for increased action on climate change, and this movement has spread to 185 countries and received widespread media attention. This exploratory study investigates motives for participating or not in the climate strikes and future participation among students in Switzerland. In a sample of $N = 638$ university students, we found that trust in climate scientists, low trust in governments, response efficacy, protest enjoyment and the perceived success of the strikes predicted participation. Contrary to statements in the public media but consistent with the literature, students who participated in the climate strikes reported consuming less meat, flying less and taking more steps to compensate the $CO_2$ emissions from flights compared to students who did not participate. We discuss how the insights from this study help reveal the determinants of youth collective action on climate change.

## Introduction

Fridays for Future is the international climate strike movement founded by Swedish activist Greta Thunberg to call for immediate action on climate change. In 185 countries, an estimated 7.6 million people have attended Fridays for Future climate strikes to raise awareness of the threats posed by climate change and to place pressure on political parties to act on climate change [1]. The movement has received widespread media attention and its call to action is supported by thousands of scientists [2]. The Fridays for Future strikes are one of the largest environmental youth movements up to date, which raises the question of what drives so many students all over world to take part in this movement. Understanding which factors influence the youth to engage or not in these strikes could provide valuable insights into how and why current and future generations engage in collective action on climate change. Therefore, we explore the determinants that influenced students to take part or not take part in the Fridays for Future strikes, as well as the factors that influence future intended participation. Understanding the drivers of young people to take part in the Fridays for Future strikes can further

**Data Availability Statement:** Most relevant data are within the paper and its Supporting Information files. Additionally, the dataset used to compute analyses is available here: https://osf.io/e5uby/.

**Funding:** The authors received no specific funding for this work.

**Competing interests:** The authors have declared that no competing interests exist.

provide more insights into youth activism and collective action, which has been shown to differ from adult activism and collective action [3]. Where possible, we therefore focused on youth participation literature.

Previous research readily describes youth activism and youth movements but comes to different conclusions regarding the level of political participation of young people. Some studies find that people under the age of 25 are politically apathetic and disengaged [4–6]. For example, by looking at data from eight industrialised countries collected over the period 1993–2010, Dalton [7] found that participation in environmental protests by young people under the age of 30 fell by 50% during that period. However, other studies indicate that young people do not have less interest in political topics than the rest of the population, but are instead using different practices of political engagement (e.g. boycotts and buycotts [8]) compared to mainstream practices such as protests participation [9–11]. In general, young people seem to be turning to different ways of political engagement as they feel ignored and excluded by the current political scene, and express low trust in politicians and the political party system [9, 12]. This low trust might lead to stand-by citizens who are interested in political topics but only become active once the circumstances call for it [11]. The Fridays for Future strikes might present such a circumstance that enabled young people to become politically active through protest participation on the topic of climate change, which is a topic which youth are consistently concerned about [13].

Perhaps due to the perceived political disengagement of the youth in certain political mainstream structures, some media outlets have questioned the genuine intention of the Fridays for Future strikers [14, 15] and called their behaviour "inconsequential" [14]. Understanding the drivers that led some students to participate at these strikes is thus interesting not only because of the global outreach of the movement, but also because these strikes mark a shift in political engagement and protest participation by the youth.

The current study advances the literature on collective action and protest participation in several ways. First, it provides some of the first empirical evidence on the motives for participation in the Fridays for Future strikes, one of the largest environmental social movements to date (for other studies on the climate strikes see: [16–20]. Second, it adds to the current literature on climate protest participation due to its novel approach of investigating the drivers of both participation and non-participation. Indeed, social movement studies have so far neglected research on individuals choosing not to participate in protests. We are only aware of one other study that compared participants with non-participants at environmental protests [16]. The lack of research on non-participation was recently noted in a review [10]. As van Stekelenburg and Klandermans [21] summarise, comprehension of protest participation is about trying to understand why certain people that are in the seemingly same situation respond differently to it, e.g., some protest in the streets for climate action while others do not. Third, while previous studies investigated socio-psychological models of protest participation based on the intention to participate [22], our study assesses the determinants of both participation and intended future participation. Exploring similarities and differences in the factors predicting participation and future intended protest participation can provide insights on the intention-behaviour gap in the context of environmental protests. Fourth, previous studies on protest participation have largely relied on samples of the general population. However, the participation motives of younger individuals might differ from older individuals [16]. Taking up Ostrom's [23] call to investigate behavioural theories of collective action, we aim to develop a better understanding of the psychological mechanisms that led students to participate in the Fridays for Future strikes and, in doing so, to extend the literature on collective action.

## The climate strikes as collective action

While individual actions can help to address the global climate crisis, for example through changes in private consumption, several authors questioned the efficacy of individual behaviours and called for collective and system-wide actions to halt climate change [22, 24]. This has led to an increasing amount of research dedicated to the analysis of determinants of environmental collective action [25, 26]. The current study seeks to advance knowledge on collective action by considering a large phenomenon of collective action on climate change, i.e., the Fridays for Future climate strikes. Given the lack of a unified theoretical foundation for European youth political participation [10] and only few existing empirical and peer-reviewed investigations on the climate strikes (e.g. 16), we will review the most important determinants for engagement in collective action among youth.

Studies on the climate strikes show that participants are largely driven by strong concern and worry about climate change [16, 18]. This is in line with theoretical arguments from the wider collective action literature [27, 28]. Namely, as theorised and empirically confirmed by van Zomeren et al. [29, 30], individuals can cope with climate change through an emotion-based and/or a problem-based approach. Both approaches serve to predict the intention to participate in environmental protests. When faced with the reality of climate change, some individuals experience fear of possible negative consequences. On the one hand, an emotion-focused type of coping is based on the regulation of these emotions [31], with more concerned individuals being more likely to take action to reduce the negative consequences they fear [30]. On the other hand, collective efficacy beliefs represent a problem-focused coping strategy, whereby individuals engage in collective action when they perceive the group to be able to solve a collective action problem (19). Based on the literature, participation in the Fridays for Future strikes will likely be determined by climate change worry and perceived collective efficacy. Indeed, van Zomeren et al. [30] find that both emotion-focused and problem-focused coping approaches explain environmental action. However, this problem-based approach has recently been questioned, as negative emotions (e.g. anger) and positive emotions (e.g. joy) have been shown to mediate the effect between collective efficacy and collective action [32].

While a significant amount of literature has highlighted the role of negative emotions (e.g., anger, worry, depression, anxiety, grievances, rage) for collective action [21, 30–33] and climate activism more specifically [16, 34], fewer studies have considered the role of positive emotions (e.g., joy, pride) as predictors of collective action [32, 35, 36]. Given that the early evidence on climate strike participation showed that both positive and negative emotions influenced participation [16, 18], we assessed both climate change worry (a negative emotion) and protest enjoyment (a positive emotion).

As postulated by the problem-focused approach [30], and as highlighted in several models of collective action [22, 26, 27, 37], collective efficacy, or the expectation that one's group is able to achieve social change through collective action [38], is a strong and consistent predictor of collective action and climate strike participation [16]. Similar to collective efficacy, the perceived success of a protest, the belief that the protest will be successful in achieving its goal, predicts protest participation [39]. Next to perceptions of collective efficacy, perceptions of self-efficacy and response efficacy have also been analysed in the context of environmental activism and protest participation. While self-efficacy refers to the belief that one is capable of performing an action, response efficacy refers to the belief that one's actions to achieve a certain goal will be effective [33]. Response efficacy has therefore also been termed goal efficacy [25]. Hamman and Reese [25] further differentiated between direct and indirect goal/response efficacy, with direct response efficacy relating to the efficacy of one's own actions to achieve a goal and indirect response efficacy relating to one's efficacy to influence or encourage others

to achieve a goal. In the case of climate strike participation, we argue that individuals will be more likely to participate at a climate strike if they believe that their actions (e.g. strike participation) will influence others to also act to mitigate climate change, i.e. they expect that others will increase action as a response to their own action. Indeed, indirect response efficacy has been found to positively predict activist pro-environmental behaviour [25]. Theoretical reasoning and empirical evidence both point towards the importance of collective and response efficacy for collective action [22, 25, 33, 40], but the importance of self-efficacy beliefs is somewhat contested. Namely, while some authors argue that self-efficacy is not relevant to problem-focused coping and environmental behaviour [30, 38, 41], other authors find self-efficacy to predict environmental protest behaviour [42–44]. We measured participants and non-participants' perception of perceived success of the strikes as well as collective, response and self-efficacy beliefs with regards to climate change.

Recent work has shown the importance of trust in climate scientists for the uptake of pro-environmental behaviours, climate policy support and environmental civic actions [45, 46]. We believe the analysis of trust in climate scientists as a predictor of climate strike participation to be important for several reasons. First, the main claim of the Fridays for Future strikes is 'listen to the scientists', which indicates that trust in scientists is a core element of this youth movement [47]. Second, trust in different societal actors (including scientists and politicians), is assumed to create positive emotions of hope that help the young face climate change concerns constructively [48]. For example, by conducting interviews with young people engaged in sustainability-focused organisations, Ojala [49] found that trust in science made interview participants hopeful and fuelled their engagement. Yet, the analysis of trust in scientists for youth engagement remains understudied and recent calls have been made to increase knowledge on how trust in scientists affects youth engagement [48]. While there are strong indications for trust in climate scientists to positively predict participation in climate strikes, the role of trust in political institutions and politicians is not so clear. Previous research found positive associations between trust in political institutions/governments and pro-environmental behaviours and environmental civic actions [45, 46]. However, from a theoretical point of view, if the youth trusted political institutions to successfully address climate change, this would preclude the need to protest in the first place. In line with this reasoning, Finkel and Müller [39], argued that discontent with the provision of public goods by governments is an important precondition for political protesting. The first surveys on German Fridays for Future strikers showed that strikers had low trust in policymakers to address climate change [50]. Further, an analysis of tweets with the hashtag #SchoolStrike4Climate found that 14% of tweets attacked or blamed the government for their inaction on climate change and reflected distrust in future promises of government action [51]. This is in line with research on youth protest participation finding low trust in politicians and the political system [9]. We also assessed how trust in political institutions to handle the climate crisis affected participation and intended future participation in the Fridays for Future climate strikes.

As the Fridays for Future participants have been challenged in the (Swiss) media and criticised for their alleged unwillingness to make personal behaviour changes [14, 15], we further analysed whether students who participated in the strikes reported more climate-friendly behaviours than non-participating students. A recent qualitative study seems to dispute these accusations [18], finding that the majority of participants at climate strikes declared having made significant lifestyle changes, including a reduction in meat consumption. Indeed, consumer politics (i.e. boycotting and buycotting) reflects a practise of political engagement by the youth [8]. Differences in consumer behaviour could therefore also reflect a form of political engagement. Finally, we assessed participants' gender and political orientation, as being a woman and politically progressive both predict more environmental activism [16, 42].

## Methods

### Participants

To recruit a sample that included both university students that did and did not participate in the climate strikes, teaching staff at ETH Zurich and the University of Zurich (Switzerland) gave permission to conduct the survey during lectures. When compared to other methods such as sharing the survey link via email, this approach allowed us to reduce the likelihood of self-selection bias, whereby only those people who were interested in the climate strikes would complete the survey. Overall, 34 lecturers from different departments were contacted, of whom 12 allowed us to conduct the survey during lectures. Students were given brief instructions as to how to fill out the voluntary survey. In total, 638 students completed the survey, of whom 30.3% had attended at least one of the climate strikes. The students in our sample had an average age of 23 years (*SD* = 3.73), 52% were women and most identified as being on the political left (55.7% left wing, 36.1% political centre and 8.2% right wing). Written consent was obtained from all participants. This study was approved by the Ethics Commission Office at ETH Zurich, EK 2019-N-49.

### Measures

**Climate change worry.** We developed a new scale for measuring specific worries about climate change (Cronbach's alpha = .78). We asked respondents to state how worried they were about climate change having a strong impact on sea-level rise; the loss of biodiversity; health-related risks; precipitation patterns; and extreme weather events and human migration, each on a six-point Likert scale (1 = *not worried at all* to 6 = *very worried*).

**Trust in government.** Two items measured trust to act on climate change: in (1) the world's governments and (2) in the Swiss government, on a six-point Likert scale (1 = *do not trust at all* to 6 = *trust completely)*. Reliability for this scale was good, with Cronbach's alpha = .75.

**Trust in climate scientists.** A single item asked respondents how much they trusted climate scientists to provide correct information on climate change, on a six-point Likert scale (1 = *do not trust at all* to 6 = *trust completely*).

**Self-efficacy.** One item was adapted from Marquart-Pyatt [44]: 'It is just too difficult for someone like me to do much about climate change' on a six-point Likert scale (1 = *strongly disagree* to 6 = *strongly agree*). This item was reversed for analysis.

**Response efficacy.** The following item was used to measure response efficacy on a six-point Likert scale (1 = strongly disagree to 6 = strongly agree): 'My actions will influence others to behave in ways that mitigate the effects of climate change.'

**Collective efficacy.** One item: 'Humans can reduce climate change and we are going to do so successfully' was rated on a six-point Likert scale (1 = *strongly disagree* to 6 = *strongly agree*).

**Protest enjoyment.** To measure the extent to which the participants enjoy participating in protests in general, we created a protest enjoyment scale comprising three items on a six-point Likert scale (1 = *strongly disagree* to 6 = *strongly agree*). More specifically, we asked participants how much they generally enjoy taking to the streets to protest, if they enjoy being part of a movement and if they enjoy challenging the established order (Cronbach's alpha = .70).

**Perceived success.** To measure the perceived success of the climate strikes in terms of making a difference with regards to climate change, we used a single item with a six-point Likert scale (1 = *not successful at all* to 6 = *very successful*).

To assess climate-friendly behaviour, we asked participants how often they eat meat per week (0 = *never* to 6 = *more than five times)*, how many times they had flown in 2017 and 2018 (0 = *never* to 6 = *more than five times)* and how many of those flights had been intercontinental flights (0 = *none* to 5 = *more than four*). We also asked those participants who reported having flown at least once during the period in question whether they had taken steps to offset their flight emissions. Finally, we also asked participants whether they had donated money to an environmental organisation within the past year.

To assess participation in the strikes, we asked participants if they had ever participated in a climate strike, if they had participated more than once and if they were planning to participate in the next strike. The first two questions were dichotomous (1 = *no*, 2 = *yes*), while participants had the option to state 'Don't know' in response to the third question. Participation at the strikes and intended future strike participation were used as dependent variables in our linear regression models. Last, we asked respondents to indicate their age, gender, current level of study, field of study, and political orientation. Participants' knowledge about climate change was assessed but not included in the analyses due to insufficient reliability (S1 File). A summary of the scales used for analyses as well as the entire questionnaire can be found in S1 File.

### Data analysis

First, we conducted explanatory factor analyses for our climate change worry and participation enjoyment scales to assess their validity. Then we conducted analyses to understand how students that did take part in the strike(s) differed from students who did not take part, as well as to reveal predictors of participation. We used Welch's two sample t-tests to investigate differences in climate change concern, trust in governments and in climate scientists, self-efficacy, response efficacy, collective efficacy, protest enjoyment, perceived success and pro-environmental behaviour between the participants who attended the climate strikes and those who did not. To examine the differences in $CO_2$ compensation, the donation of money to environmental organisations and gender, we used chi-square analyses. To analyse the predictors of participation and future intended participation in the climate strikes, we used logistic regression models. Multicollinearity of variables was assessed with the variance inflation factor. Based on our review of the literature, we included the following variables in our model to predict participation and future intended participation in the climate strikes: self-efficacy, response efficacy, collective efficacy, trust in governments, trust in climate scientists, perceived success of the strikes, (overall) climate change worry, protest enjoyment, gender and political orientation. All analyses were conducted in R Studio [52]. Additionally, the code and dataset used for these analyses can be found here: osf.io/e5uby.

## Results

### Differences between participants and non-participants

To assess the validity of our climate change worry scale, we verified sampling accuracy with the Kaiser-Meyer-Olkin (KMO) measure and it resulted in a value of 0.8 (above the commonly recommended value of 0.60). Bartlett's test of sphericity further indicated sufficient significant correlation in the data for factor analysis, $\chi^2 (15) = 1014.74$, $p < .001$. A scree plot indicated that only one component had an eigenvalue larger than 1, supporting our approach. Further, all items had factor loadings $> 0.60$ except for worry about extreme weather events (0.45) and human migration (0.49). The scale also had an acceptable Cronbach's alpha of .78. Regarding our protest enjoyment scale, we again verified sampling accuracy with the Kaiser-Meyer-Olkin (KMO) measure which resulted in a value of 0.6. Bartlett's test of sphericity further indicated sufficient significant correlation in the data for factor analysis, $\chi^2 (3) = 421.08$, $p < .001$. A factor analysis showed that the first two items had adequate factor loadings (Item 1 = 0.76, Item

**Table 1. Descriptives of the predictors and outcomes of Welch's two sample t-tests between strikers and non-strikers.**

| | Participation ($n$ = 193) | No participation ($n$ = 443) | Difference |
|---|---|---|---|
| | Mean (SD) | Mean (SD) | |
| Climate change worry (overall) | 5.21 (.63) | 4.63 (.87) | $t$ (490) = 9.5*** |
| *Specific climate change worries* | | | |
| Sea-level rise | 5.18 (.90) | 4.54 (1.24) | $t$ (497) = 7.33*** |
| Loss of biodiversity | 5.62 (.85) | 5.04 (1.12) | $t$ (477) = 7.24*** |
| Health-related risks | 4.72 (1.23) | 4.34 (1.34) | $t$ (396) = 3.46*** |
| Precipitation patterns | 5.23 (1.04) | 4.61 (1.24) | $t$ (433) = 6.49*** |
| Extreme weather events | 5.55 (.80) | 5.07 (1.08) | $t$ (483) = 6.23*** |
| Human migration | 4.96 (1.31) | 4.19 (1.47) | $t$ (404) = 6.52*** |
| Trust in governments | 2.65 (.89) | 2.98 (.95) | $t$ (387) = 4.28*** |
| Trust in climate scientists | 5.23 (.77) | 4.71 (.99) | $t$ (464) = 7.16*** |
| Self-efficacy | 4.26 (1.29) | 3.90 (1.31) | $t$ (371) = 3.23** |
| Response efficacy | 4.51 (1.11) | 3.59 (1.2) | $t$ (391) = 9.33*** |
| Collective efficacy | 3.18 (1.14) | 3.04 (1.1) | $t$ (351) = 1.45 |
| Protest enjoyment | 4.18 (1.1) | 2.87 (.92) | $t$ (313) = 14.46*** |
| Success | 4.37 (.97) | 3.17 (1.18) | $t$ (439) = 13.34*** |
| Meat consumption | 2.35 (1.63) | 4.18 (1.85) | $t$ (408) = 12.43*** |
| Flights | 3.49 (1.93) | 4.12 (2.11) | $t$ (396) = 3.69*** |
| Intercontinental flights | .96 (1.26) | 1.14 (1.50) | $t$ (422) = 1.48 |
| $CO_2$ compensation | 1.40 (.49) | 1.22 (.41) | $\chi^2$(1) = 18.69*** |
| Donating money | 1.46 (.50) | 1.19 (.39) | $\chi^2$(1) = 47.98*** |
| Male | 1.39 (.49) | 1.52 (.50) | $\chi^2$(1) = 8.021** |
| Political orientation | 1.85 (.74) | 2.60 (.77) | $t$ (375) = 11.49*** |

Note. Reference categories: $CO_2$ compensation: 1 = no, 2 = yes; Donate money: 1 = no, 2 = yes

**$p$ < .01

***$p$ < .001.

2 = 0.84), whereas Item 3 resulted in a factor loading of 0.41. However, given the conceptual similarity of the items and an acceptable Cronbach's alpha of .70, this scale was also included.

We found that individuals who attended the strikes were more worried about climate change than individuals who did not attend them (Table 1). In fact, students who participated in the strikes reported being more worried about sea-level rise, the loss of biodiversity, health-related risks, precipitation patterns, the frequency and magnitude of extreme weather events and human migration than students who did not participate. In line with previous literature [53], among the climate risks presented in the survey, the current sample was least worried about health-related risks.

Our findings indicate that students who participated at the strikes had less trust in governments to act on climate change and more trust in climate scientists than non-participants. Students who attended the strikes reported more self-efficacy and response efficacy but no difference in collective efficacy compared to non-participants. Moreover, students who attended the strikes enjoyed protesting more and perceived the strikes to be more successful than the students who did not attend the strikes.

### Pro-environmental behaviour

Students who attended the strikes reported eating significantly less meat, with 47% of the participating students declaring that they never eat meat compared to 11% of the non-

participating students. Students who attended the strikes also reported flying significantly less frequently than the students who did not attend the strikes, although for both groups only 13% of participants reported not having flown in the last two years. Interestingly, regarding inter-continental flights, we found no significant difference between the two groups. Of the participants who reported having taken a flight at least once in the last two years (87%), significantly more students (40%) who participated in the climate strikes reported having taken steps to compensate their $CO_2$ emissions than students who did not participate (21%). Further, 46% of the students who attended the strikes reported having donated money to an environmental organisation in the past year compared to 19% of the non-striking students. Strike participation and intention to participate in future strikes strongly correlated, $r(447) = .74$, $p = <0.01$, indicating that participants who already participated at one strike are likely to participate in future strikes. In sum, students who attended the climate strikes reported engaging in more climate-friendly behaviours than non-attending students. Regarding socio-demographics, more women attended the climate strikes than men, and the striking students were more politically left-wing than the non-striking students.

## Predictors of participation and future intended participation

In line with the literature and our hypotheses, lower trust in governments and higher perceived success of the strikes predicted strike participation. Collective efficacy, however, did not predict strike participation and indeed, we found no correlation at all between collective efficacy and strike participation (Table 2). Similarly to collective efficacy, self-efficacy was weakly correlated with participation but was not a unique predictor in the regression. However, response efficacy positively predicted participation, meaning that the more participants believed that their actions could have a positive influence on others, the more likely they were to have participated in a strike. Interestingly, response efficacy did not predict future intended participation (Table 3). In line with previous research [45], trusting climate scientists to provide correct information on climate change predicted strike participation.

**Table 2. Correlations of key variables (N = 638 for participation and N = 446 for intended future participation).**

| Variables | 1 | 2 | 3 | 4 | 5 | 6 | 7 | 8 | 9 | 10 | 11 |
|---|---|---|---|---|---|---|---|---|---|---|---|
| 1. Participation | | | | | | | | | | | |
| 2. Intended participation | .74** | | | | | | | | | | |
| 3. Worry | .33** | .43** | | | | | | | | | |
| 4. Self-efficacy | .13** | .10* | .15** | | | | | | | | |
| 5. Response efficacy | .35** | .42** | .31** | .33** | | | | | | | |
| 6. Collective efficacy | .05 | .14** | .09* | .04 | .17** | | | | | | |
| 7. Trust governments | -.16** | -.12* | -.13** | -.03 | .01 | .25** | | | | | |
| 8. Trust scientists | .26** | .25** | .31** | -.02 | .17** | .13** | .09* | | | | |
| 9. Protest enjoyment | .51** | .62** | .37** | .12** | .33** | .13** | -.20** | .20** | | | |
| 10. Perceived success | .45** | .59** | .41** | .12** | .38** | .23** | .02 | .30** | .41** | | |
| 11. Male | -.11** | -.15** | -.11** | -.09* | -.08* | .05 | -.07 | .13** | -.04 | -.12** | |
| 12. Political orientation | -.42** | -.52** | -.32** | -.12** | -.29** | -.01 | .21** | -.24** | -.46** | -.34** | .09* |

Note. Point-biseral correlations were used to calculate the relationship between participation/intended participation/gender (male) and all other variables. Phi coefficients were computed to show the relation between gender (male) and participation/intended participation. All other correlations were computed with Pearson correlation coefficients.

*$p < .05$.

**$p < .01$.

**Table 3. Logistic regression analyses explaining participation and future intended participation in climate strikes.**

| Predictors | Participation | | | Future intended participation | | |
|---|---|---|---|---|---|---|
| | Odds Ratio | 95% CI | *p* | Odds Ratio | 95% CI | *p* |
| (Intercept) | 0.00 | [0.00 – 0.02] | < .001 | 0.00 | [0.00 – 0.01] | < .001 |
| Self-efficacy | 1.05 | [0.87 – 1.27] | .604 | 0.78 | [0.60 – 1.01] | .065 |
| Response efficacy | 1.34 | [1.06 – 1.70] | **.014** | 1.23 | [0.89 – 1.70] | .211 |
| Collective efficacy | 0.84 | [0.68 – 1.05] | .135 | 0.90 | [0.67 – 1.21] | .487 |
| Trust governments | 0.72 | [0.55 – 0.94] | **.016** | 1.04 | [0.72 – 1.50] | .834 |
| Trust scientists | 1.46 | [1.10 – 1.97] | **.010** | 0.88 | [0.60 – 1.28] | .497 |
| Perceived success | 2.06 | [1.60 – 2.69] | **< .001** | 2.83 | [2.03 – 4.08] | **< .001** |
| Climate change concern | 1.02 | [0.72 – 1.46] | .902 | 1.59 | [1.00 – 2.60] | .057 |
| Protest enjoyment | 2.33 | [1.82 – 3.02] | **< .001** | 3.30 | [2.34 – 4.80] | **< .001** |
| Men | 0.61 | [0.38 – 0.97] | **.040** | 0.65 | [0.35 – 1.22] | .183 |
| Political orientation | 0.58 | [0.42 – 0.81] | **.002** | 0.47 | [0.29 – 0.74] | **.002** |
| | $R^2$ Tjur = 43.6%, N = 631 | | | $R^2$ Tjur = 59.9%, N = 445 | | |

Note. $^*p < .05$, $^{**}p < .01$, $^{***}p < .001$.

Interestingly, climate change worry did not predict participation in the regression, even though the two variables were strongly correlated, $r(633) = .32$, $p = <0.01$; see (41). However, climate change worry was closer to the alpha threshold when predicting intended future participation. Perceived success and protest enjoyment were the strongest predictors of strike participation, while women and left-wing oriented students were also more likely to have participated at a strike. Thus, positive emotions seem to be stronger predictors of participation than negative emotions. When comparing participation and future intended strike participation, response efficacy, low trust in governments, trust in scientists and gender only predicted participation but not intended strike participation, and protest enjoyment and perceived success of the strikes remained the strongest predictors also for future intended participation.

## Discussion

The Fridays for Future climate strikes have been the subject of considerable media attention, with students striking in 2,376 cities across 135 countries in March 2019 [1]. Based on the wider literature on collective action and first empirical studies on the climate strikes, this exploratory study provides initial empirical evidence for the motives that led students to participate or not participate in the climate strikes in Switzerland.

The results of our study are partially in line with the wider literature on collective action. Namely, we found that low trust in governments and perceived success of the strikes predicted strike participation, and perceived success further strongly predicted future intended participation. Thus, the youth might be more inclined to participate in the strikes if they believe that the governments' current efforts to address the climate crisis are insufficient and if they perceive the strikes to have the potential to successfully address climate change. This reflects the findings by Boulianne et al. [51] who found tweets by climate strikers to show low trust in governments to address climate change. As expected given previous findings [45, 46], the more students trusted climate scientists to provide correct information on climate change, the more likely they were to have participated at a strike. As one of the main claims of the Fridays for Future movement is for politicians and societal actors to 'listen to science', future studies could investigate determinants of trust in climate scientists among the youth, as called for by Ojala [48]. Interestingly, trusting scientists did not predict future intended strike participation. The

different predictors of participation and future strike participation as well as the differing predictive strengths of the two models highlight the importance of comparing both participation and intended future strike participation.

We did not fully replicate previous findings related to collective efficacy [16, 30, 39], as collective efficacy was not correlated with participation, therefore not providing support for the problem-focused coping approach [30]. We ascribe this finding mainly to the item used to assess collective efficacy. Namely, we did not measure perceptions of collective efficacy specifically related to the Fridays for Futures strikes, but measured collective efficacy related to climate change in general. A recent study on the Fridays for Future strikes in Germany also found collective efficacy not to be a significant predictor of participation in their linear model [20] (although the authors find collective efficacy to correlate with activism). Interestingly, the authors also assessed collective efficacy with only one item related to climate change in general and not specifically to the protests. A further issue might be related to the wording of our item, as we asked participants whether humans can reduce climate change and are going to do so successfully. The issue at hand may lie in the word 'successfully'. For example, participants might have perceived that humans can collectively act on climate change, but the definition of successful might be different for different individuals. Further, contrary to our expectations, collective efficacy tended towards negatively predicting participation, even though not significantly. This result is somewhat in line with the results by Hamman and Reese [25], who found collective efficacy beliefs in humanity to address climate change (similar to our item) negatively predicted activist behaviour. The authors explain this finding by referring to Olson's [54] paradox that collective efficacy could potentially lead to inaction if a single member's behaviour is perceived as unnecessary for goal achievement. For all these reasons, we discourage future use of our item to assess collective efficacy and advise future research to assess collective efficacy specifically related to the climate strikes and not just to climate change in general.

Contrary to other findings [42–44], self-efficacy did not predict participation. Even though levels of self-efficacy were higher for students that attended at least one strike (Table 1), this did not translate into self-efficacy predicting protest participation. While self-efficacy is an important predictor for environmental activism in general, we now believe that response efficacy might be more important in predicting participation at a global protest. Students might not only participate in a strike because they believe that they as single individuals can make a difference, but because they believe that their action (participating at a protest) can influence the actions of others thus creating a cascading effect. This is in line with the findings by Hamann and Reese [25] who found that while (indirect) response efficacy predicted activist pro-environmental behaviours, self-efficacy did not. Additionally, self-efficacy might play a stronger role in behaviours that are more challenging to perform. Attending the climate strikes was a relatively easy behaviour to perform for the individuals included in the sample of this study, as they were all students attending the university where the protest took place. In other words, the students only needed to leave the building to attend the strikes. As most students likely believed that they were able to perform this action, it is possible that self-efficacy had lower predictive power for strike participation in this context, compared to performing pro-environmental behaviours that were more challenging to perform, such as changing transportation mode or donating money.

A further explanation for why response efficacy might be more relevant for protest participation than self-efficacy is grounded in the work of Finkel et al. [37], who showed that individuals engage in collective action when they perceive that the group will only succeed through the participation of all its members, which they termed a perception of unity. Thus, students with higher response efficacy perceptions might be more likely to join the protests as they

might feel that their own participation can influence others to join the strikes. This reasoning is in line with the work by van Zomeren et al. [28] who found participatory efficacy (i.e. the belief that one's participation in a strike will contribute to the overall success of a strike) to be a distinct type of efficacy that represents a unique predictor of collective action. Therefore, we recommend future studies assess participatory efficacy alongside collective efficacy and self-efficacy. Additionally, Finkel and colleagues [37] argued that perceptions concerning the necessity of group unity are often coupled with perceptions of a moral duty to participate in the protests. Previous research investigating the motives for participation in the 2014 New York City People's Climate March through qualitative interviews confirms the validity of this explanation [55]. While some of those participants felt doubtful as to their personal ability to reduce climate change (low self-efficacy perceptions), they nevertheless took part in the protest out of a sense of duty and responsibility. Moreover, participants declared that they had felt hopeless regarding climate change until they saw the large crowds at the march, which gave them a sense of hope that climate change could be reduced through their participation. Constructivist hope, or the feeling that individuals or collectives of people can reduce climate change, predicts political environmental behaviour [56]. Based on the literature, we argue that even though perceptions of self-efficacy did not predict participation in our study, students might have participated in the climate strikes because they a) perceived the strikes to be successful in making a difference, b) perceived that their participation would positively influence the actions of others, and/or c) acted out of a sense of duty and responsibility. Perceived duty and sense of hope are two constructs that might have strong influences on protest participation, but they are still largely absent from the literature. Thus, we encourage future researchers to include them.

As postulated by several studies [21, 33], we found the negative emotion of climate change worry was strongly, positively correlated with participation and intended future strike participation, even though climate change worry did not predict participation and future strike participation in a regression with many collinear terms. We ascribe this to the fact that climate change worry correlates with other determinants of participation such as perceived success and protest enjoyment. In line with findings of Opp [35] and a growing literature on the role of positive emotions [32, 36], we further show that protest enjoyment was the strongest predictor of strike participation. Protest enjoyment has been depicted negatively in the media, with students being portrayed as protesting 'just for fun' and for skipping school. One item in our protest enjoyment scale asked how much students enjoy challenging the established order, while another item assessed how much students like being part of a movement. Thus, we do not interpret our results as indicating that students only participated for fun (and caution other researchers not to do so), but we suggest that general enjoyment of being part of movements and protesting are important predictors for both participation and future intended strike participation. Indeed, protest enjoyment is the strongest predictor in both models. Our results support previous research findings that positive emotions such as enjoyment also predict protest participation [32] and we encourage future research to further investigate the role of positive emotions for climate activism. However, most prior studies focused on the pleasure and enjoyment that result from engaging in environmental behaviour rather than analysing enjoyment as an antecedent of environmental behaviour [57, 58]. Our findings support the idea that making environmental actions appear more enjoyable could potentially motivate more people towards such actions. Future research could analyse the relationship between enjoyment and environmental action in order to foster a better understanding of how joy influences environmental behaviour, as well as how joy can be used to stimulate environmental behaviour. Lastly, enjoyment of being part of a movement can loosely also be interpreted as a variable measuring social identity. Social identity (i.e., the identification with a group) also

strongly predicts engagement in environmental protest [26] and more specifically, identification with climate strikers predicts participation in the climate strikes [16]. The fact that we did not assess social identity presents a strong limitation of this study and we recommend future study to assess this key variable.

We also found that political orientation and protest enjoyment were strongly negatively correlated, which indicates that left-wing participants generally enjoyed protesting more than right-wing participants. Right-wing participants further exhibited more trust in the world's governments and in the Swiss government to act on climate change than left-wing participants, which may reflect ascription of responsibility as well as trust. We also found a negative correlation between political orientation and trust in scientists, which indicates that right-wing participants trusted climate scientists to provide accurate information concerning climate change less than the left-wing participants. This is in line with previous findings that trust in climate science is politically polarised [59]. These findings provide valuable insights into the motives for (and for not) participating in the climate strikes based on an individual's political orientation.

Regarding climate-friendly behaviours, we found that individuals who participated in the climate strikes reported more climate-friendly behaviours than individuals who have never attended a climate strike. Students who participated in strikes have been challenged in the Swiss media and criticised for their alleged unwillingness to make personal behaviour changes [14, 15]. However, our findings appear to counter such accusations. Interestingly, while the striking students declared eating less meat per week, having flown less in the past two years and having sought to offset the $CO_2$ emissions from their flights more often than non-striking students, we found no difference in long-haul flights. One possible reason for this might be that it is easier to find alternative modes of transport within Europe than for oversee destinations. Future research could use a longitudinal approach to examine whether to what extent continued participation in the climate strikes influences individual pro-environmental behaviour, as well as whether the pro-environmental behaviour of striking students has spill-over effects on other members of society.

Last, our results provide insights for marketing and activism strategies. For example, evoking feelings of response efficacy could be an important strategy to increase participation at the strikes. This could be done by using specific climate change imagery [60]. Second, as the two groups strongly differed in their levels of general protest enjoyment, the climate strikes could be made more attractive to a larger audience by considering different individual factors that influence protest enjoyment. This relates to Klandermans' [61] notion that protest participation not only depends on the demand for a protest but also on its supply: for example, the opportunities staged by strike organisers. The organisers should thus be aware of individual differences in protest enjoyment and protest demand and supply a wider action repertoire for people with different individual motivations to participate in the climate strikes. Since participating and non-participating students differ in their perception of success of the strikes, a third strategy could be greater communication of achieved successes by the climate strike movement. Future research could expand on these findings by considering additional factors that influence protest participation such as investigating which social identities people hold and in what contexts those identities promote or inhibit pro-environmental action [62].

Our study thus advances the literature on collective action in several ways. First, we show that trust in climate scientists is an important determinant for participation in the climate strikes. While the importance of trust in science and scientists for youth engagement has been acknowledged in the literature [63], this is the first study to show its importance for climate strike participation. Second, we show that low trust in governments (both national and international) to act on climate change positively predicts youth participation in climate strikes. As

previous studies found trust in political institutions to positively predict individual pro-environmental behaviours and policy support [45, 46], the opposite seems to be true for climate activism. We recommend future studies to consider the role of trust in different actors for protest participation. Third, we provide further evidence that the role of positive emotions is a strong predictor of protest participation, and in our case, an even stronger predictor than negative emotions such as worry. Fourth, we show that predictors differ between participation and future intended participation, thus shedding light on the intention-behaviour gap in strike participation.

## Limitations

First, while teaching staff instructed their students to fill out the survey and the present researchers supervised the process, we cannot assure that all the students filled out the survey. Therefore, we cannot exclude the possibility of self-selection in participation. However, we do not expect self-selection to have influenced the results of the study because the number of filled out surveys did not deviate strongly from the number of students in the classrooms.

Second, the sample only included university students, while the Fridays for Future strikes are principally attended by high-school students. Therefore, the results obtained in this study might not be generalisable to younger strike participants. However, there are no specific theoretical reasons from the current frameworks to suppose that the motivations of high-school students to take part in the strikes differ from the motivations of university students. Nevertheless, future studies concerning the climate strikes should use a more representative sample for more generalisable results. Similarly, our study only assessed factors determining participation among Swiss students and future studies could analyse whether the motives for participation differ among different countries. Additionally, we suggest that future studies concerning the climate strikes, and collective action more generally, use domain-specific measures to evaluate the influence of self-efficacy, response efficacy and collective efficacy in relation to participation, as recommended in the literature [64].

The perception of strike success was assessed after participation. Since we could not assess perceived success of the strikes prior to participation, we cannot exclude the fact that perceived success might have increased after their participation, as was the case for participants in the study by Grecni et al. [55]. However, as the strikes gained widespread media attention, it is possible that students perceived the climate strikes to be successful even before they participated in the strike. Nevertheless, to better understand the impact of expected success of the strikes on participation in the strikes, expected success would need to be measured beforehand. We expect that the perceived successfulness of the strikes will influence strike participation, as it was a strong predictor of the intention to participate in future strikes.

The behavioural outcomes such as diet and transportation were measured with self-report. Social desirability could have led to biased reports of flying less or eating less meat [65]. Participants that reported taking part in the strikes might be more inclined to give socially desirable answers, as they are more likely to identify themselves with people who value pro-environmental behaviour. Thus, it would be useful for future studies to measure objective pro-environmental behaviours.

## Conclusion

By investigating the motives for (and for not) participating in the climate strikes among a sample ($N$ = 638) of Swiss university students, we found that response efficacy, protest enjoyment, the perceived success of the strikes, trust in climate scientists, low trust in governments to act on climate change, female gender and left-wing political orientation predicted strike

participation, while only protest enjoyment, perceived success of the strikes and political orientation predicted intentions to join future strikes. These results contribute to the literature on collective action by including both participating and non-participating students and examining the predictors of participation and future intended participation. As literature on the Fridays for Future strikes is still nascent, we suggest that researchers further investigate the Fridays for Future strikes as a case study of global collective action on climate change.

## Supporting information

**S1 File.**
(DOCX)

## Acknowledgments

We thank our colleagues at ETH Zurich and the University of Zurich who allowed us to collect data during their lecture time. We further thank Michael Siegrist for his comments at the initial stages of this research project.

## Author Contributions

**Conceptualization:** Viktoria Cologna, Gea Hoogendoorn.

**Data curation:** Viktoria Cologna, Gea Hoogendoorn.

**Formal analysis:** Viktoria Cologna, Gea Hoogendoorn.

**Investigation:** Viktoria Cologna, Gea Hoogendoorn.

**Methodology:** Viktoria Cologna, Gea Hoogendoorn, Cameron Brick.

**Project administration:** Viktoria Cologna.

**Supervision:** Cameron Brick.

**Validation:** Viktoria Cologna, Gea Hoogendoorn, Cameron Brick.

**Writing – original draft:** Viktoria Cologna.

**Writing – review & editing:** Viktoria Cologna, Gea Hoogendoorn, Cameron Brick.

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
