## [Decision Letter · Decision Letter 0]

19 Mar 2021

PONE-D-20-32959

To strike or not to strike? An investigation of the determinants of strike participation at the Fridays for Future climate strikes in Switzerland

PLOS ONE

Dear Dr. Cologna,

Thank you for submitting your manuscript to PLOS ONE. Please accept my apologies for the delay in getting our decision to you. After careful consideration, we feel that it has merit but does not fully meet PLOS ONE’s publication criteria as it currently stands. Therefore, we invite you to submit a revised version of the manuscript that addresses the points raised during the review process.

Whilst all three reviewers acknowledged the importance of the topic and area described in your manuscript, all three also highlighted a number of substantial issues that should be addressed if the manuscript is to be further considered for publication. In particular, please note the comments made in relation to the theoretical underpinnings of the study made by all three reviewers. Please also be sure to address the comments made in relation to your measurement tools and construct variables. If you wish to revise your manuscript in response to the reviewer comments and resubmit, I would be happy to consider a substantially revised version.

Thanks for your submission.

We look forward to receiving your revised manuscript.

Kind regards,

Jim Uttley, Ph.D

Academic Editor

PLOS ONE

Journal Requirements:

1. 

2. 

1) Please change "female” or "male" to "woman” or "man" as appropriate, when used as a noun (see for instance https://apastyle.apa.org/style-grammar-guidelines/bias-free-language/gender).

2) Please include additional information regarding the survey or questionnaire used in the study and ensure that you have provided sufficient details that others could replicate the analyses. For instance, if you developed a questionnaire as part of this study and it is not under a copyright more restrictive than CC-BY, please include a copy, in both the original language and English, as Supporting Information.

4.  Please include a copy of Table 3 which you refer to in your text on page 14.

Reviewers' comments:

Reviewer's Responses to Questions

**Comments to the Author**

1. Is the manuscript technically sound, and do the data support the conclusions?

Reviewer #1: Yes

Reviewer #2: Partly

Reviewer #3: Yes

2. Has the statistical analysis been performed appropriately and rigorously? 

Reviewer #1: Yes

Reviewer #2: No

Reviewer #3: Yes

3. Have the authors made all data underlying the findings in their manuscript fully available?

Reviewer #1: Yes

Reviewer #2: Yes

Reviewer #3: Yes

4. Is the manuscript presented in an intelligible fashion and written in standard English?

Reviewer #1: Yes

Reviewer #2: Yes

Reviewer #3: Yes

5. Review Comments to the Author

Reviewer #1: One aim of the paper is to investigate the motives of past and intended future participation in the Friday for Future climate strikes among a sample (N = 638) of Swiss university students. The authors report that response efficacy, protest enjoyment, the perceived success of the strikes, trust in climate scientists, low trust in governments to act on climate change, female gender, and left-wing political orientation predicted strike participation, while only protest enjoyment, perceived success of the strikes and political orientation predicted intentions to join future strikes. The study’s second aim is to research the question whether there is a discrepancy between students’ participation in such collective actions like strikes and their own personal behavior. The study findings counter such accusations: Compared with the non striking students, the striking students declare eating less meat per week, having flown less in the past two years, and having sought to offset the CO2 emissions from their flights, however, there are no difference in long-haul flights. In the discussion section the authors critically discuss methodological problems of their research design and draw conclusion for future research.

Without question the paper targets an issue of great interest for the general public as well as social scientists. The paper is written in a good structured, readable way. The results concerning the personal behavior of the strike participants are convincing, however, are based on self-reported behavior, which may be biased to participants’ desire to be consistent with the behavior made salient by the survey (pro-environmental behavior).

For me the main weakness of the paper is the not convincing theoretical part: In general the authors seems to follow a rational choice approach (Opp, Finkel), however, they also frequently refer to the social identity approach of collective behavior (e.g. Van Zomeren). The authors do not reflect the relation between these rather different theories: Whereas rational choice models stress the role of personal benefits as central determinants of collective action and stress the social dilemma nature of collective action participation (discrepancy between self- and collective interests), the social identity do not see such a dilemma because the activation of a social identity automatically motivates the person to act as a group member. Without systematically reflecting and integrating the assumptions underlying the two different theoretical frameworks, for me it seems rather eclectic to draw single constructs from different theoretical frameworks without clarifying their theoretical relation. From my point of view such a positivistic approach is not helpful for systematical theory development within the field of collective action.

On the empirical level the used construct measure reflect the paper’s weak theoretical reflection. For example the item “enjoy being part of a movement” could also be seen as a typical indicator of the social identity construct. In this case the empirical results would indicate that social identity is the main predictor of collective action participation. A result replicating the findings of dozens of studies conducted within the social identity framework (see e.g. the work of van Zomeren). Also the discussion of the concepts of self- and response efficacy is not in line with the current theoretical discussion, I would recommend the authors to read the work of van Zomeren on the new concept of participatory efficacy (e.g., Van Zomeren, M., Saguy, T., & Schellhaas, F. M. (2013). Believing in “making a difference” to collective efforts: Participative efficacy beliefs as a unique predictor of collective action. Group Processes & Intergroup Relations, 16(5), 618-634).

Reviewer #2: This manuscript addresses a timely, interesting and still under-researched topic. The purpose and significance of the study are clearly stated in the introduction section. The authors provide sufficient information to understand the study goals and the approach followed, and a sufficient and detailed discussion is provided. Despite its potential, I believe this manuscript, at least in its current form, is not ready for publication. My major issues with it are related to the theoretical section and with some methodological limitations. I explain these issues below:

1. While the manuscript addresses some relevant literature, there is an overall trend towards selectiveness and to superficially discuss theoretical models and approaches. I believe this is misleading and a simplification in at least two ways. First, while there is some empirical evidence suggesting that young people – especially when compared to previous generations (see Maria Teresa Grasso’s work) – are more disengaged from conventional and traditional forms of political participation (e.g., voting) there is no consensus in the literature regarding other forms of participation. The myth of youth apathy has been amply discussed in the literature and it has evolved towards a more comprehensive view of the forms, patterns and levels of young people’s participation. The way previous literature is presented in this manuscript suggests a consensual view of young people as apathetic and politically disengaged, but that does not correspond to the current debates in the literature. Secondly, the manuscript also suggests a view of young people's political participation as a matter of being politically active or inactive. Such a dichotomy is again a simplification considering that previous studies have shown that even a passive stance can actually mask a latent political involvement. I strongly recommend further engagement with current debates on the field of youth participation. See for example:

- Amnå, E., & Ekman, J. (2014). Standby citizens: Diverse faces of political passivity. European Political Science Review,6(2), 261–281. https://doi.org/10.1017/S175577391300009X

- Cammaerts, B., Bruter, M., Banaji, S., Harrison, S., & Anstead, N. (2014). The myth of youth apathy: Young Europeans’ critical attitudes toward democratic life. American Behavioral Scientist, 58(5), 645–664. https://doi.org/10.1177/0002764213515992

- Barret, M. & B. Zani, B. (Eds.). Political and civic engagement: multidisciplinary perspectives (pp. pp. 499-512). Surrey: Routledge.

- Weiss, J. (2020). What Is Youth Political Participation? Literature Review on Youth Political Participation and Political Attitudes. Front. Polit. Sci., 2020. https://doi.org/10.3389/fpos.2020.00001

- Earl, J., Maher, T. V., and Elliott, T. (2017). Youth, activism, and social movements. Sociol. Compass 11, e12465. https://doi.org/10.1111/soc4.12465

2. Despite the extensive amount of literature on the topic of young people’s participation, there is still a need for studies focusing on the motives for collective action, and participation in climate strikes do mark a shift in political engagement by young people, as argued by the authors. However, this needs to be framed more carefully, considering past and current debates within the field.

3. Still related to the literature review, it was rather surprising to see that the theoretical concepts (and model) used to design the study and the hypotheses were based on the literature on collective action. Again, there are plenty of studies on the motivations for young people’s participation that could help to design and understand the findings. In addition, there is also some confusion when discussing the collective interest model, the socio-psychological collective action model, and the dual path model (in the discussion). These models, if relevant, should be better presented.

4. The authors argued that there is only one peer-reviewed study that has been conducted on the Fridays for Future strikes (line 135). I wonder why the following literature was not considered (see below) as they all were peer-reviewed and published in 2020:

- Mari Martiskainen, Stephen Axon, Benjamin K. Sovacool, Siddharth Sareen, Dylan Furszyfer Del Rio, Kayleigh Axon (2020). Contextualizing climate justice activism: Knowledge, emotions, motivations, and actions among climate strikers in six cities. Global Environmental Change. https://doi.org/10.1016/j.gloenvcha.2020.102180.

- Emilsson K, Johansson H, Wennerhag M. Frame Disputes or Frame Consensus? (2020). “Environment” or “Welfare” First Amongst Climate Strike Protesters. Sustainability. 2020; 12(3):882. https://doi.org/10.3390/su12030882

- Brügger A, Gubler M, Steentjes K, Capstick SB. (2020). Social Identity and Risk Perception Explain Participation in the Swiss Youth Climate Strikes. Sustainability. 2020; 12(24):10605. https://doi.org/10.3390/su122410605

5. There is a need for a better articulation between the empirical part and the theoretical approach. The authors should provide more details for their statistical and methodological decisions. Similarly, some of the expected results need better contextualization. For example, little is said about the role of trust and protest enjoyment in the theoretical section. Both dimensions are however key in this study. Specifically, perceived enjoyment of protest and related constructs are almost absent from the literature review section, but it is then mentioned in the discussion. I suggest introducing all the theoretical constructs in the theoretical section.

6. In my opinion, there are several strong limitations with the measures used. First, most dimensions are composed of one item (e.g., trust in climate scientists, self-efficacy, response efficacy, collective efficacy). Second, the authors created their own scales (e.g., climate change concern, protest enjoyment, pro-environmental behaviour) and there is only one reference for an item (self-efficacy item). I believe this needs to be justified, because there are already validated scales for the constructs measured. Third, while developing your own scales could be acceptable, in that case more rigorous statistical procedures should have been followed. In this regard, I would recommend to (at least) present the results of the exploratory factor analysis scales such as “climate concern scale” and “protests enjoyment”. As far as I understood, both were developed by the authors. I also wonder why pro-environmental behaviours were not considered as a composite variable for the analysis.

7. There is also an overall lack of rigour when presenting the “collective efficacy” construct and also with the way efficacy is measured. While the authors recognise the limitations of their variable “collective efficacy”, there are other aspects that should be considered. For example, group efficacy, perceived efficacy of collective action and collective efficacy are different concepts and should not be used as if they mean the same thing. In addition, considering that what was measured in collective efficacy is not even related to collective action, I would suggest avoiding claiming that “the result that collective efficacy was not a predictor of participation was surprising” or that “it doesn't fit the literature”. Probably, this finding may only represent a problem with the way the variable was measured.

8. I would recommend being more cautious in the interpretation of findings considering all the mentioned methodological limitations of this study. The contributions of the study should be more clearly articulated with the findings, and in the discussion section the authors should try to avoid just merely repeating the findings. Besides, the interpretations and suggestions made in the discussion (e.g., for example regarding the role and the multiple facets of efficacy in collective action) should be better supported by previous literature. I would also be careful in providing recommendations for future studies and claiming that there is none research on the topic. For example, there are some studies on the role of positive emotions and sense of obligation in collective action that could be used to understand the findings. As such, the recommendations for future studies should at least recognise that these already exist.

9. Implications of the research – in the introduction the authors proposed to extend the literature on collective action. This should be more clearly addressed in the final section of the manuscript.

10. Some minor issues: 1) See line 143 – “As described by Finkel and Müller 10/20/2020 8:34:00 AM”; 2) Why was prosocial behaviour considered a keyword?

Reviewer #3: In the manuscript entitled “To strike or not to strike? An investigation of the determinants of strike participation at the Fridays for Future climate strikes in Switzerland” a study with Fridays for Future protesters and non-protesters is reported. The manuscript taps at a timely topic. However, due to the following points I cannot recommend to publish the manuscript in its current form.

The theoretical framework should be clarified. For instance, the authors refer to the social identity model of collective action and derive predictions from that model (p. 5). Then they refer to the collective interest model (p. 6 f.) and derive predictions from that model. However, it is not sufficiently clear how these models and predictions are related. The authors also report that they assess trust in scientists (p. 7) and that previous research suggests that pleasure individuals get from protesting also predicts participation (p. 8) but they do not explain how they expect these variables/processes to be linked with the proposed model.

The authors introduce the distinction between emotion-focused coping and problem-focused coping, which was proposed for the context of collective action by Van Zomeren and colleagues (p. 5). However, a recent study shows that the so-called problem-focused path to collective action is mediated via emotions as well (Landmann & Rohmann, 2020). The authors may want to update their literature review including the role of positive emotions for collective action (Tausch & Becker, 2013) as well as different types of efficacy beliefs for environmental activism (Hamann & Reese, 2020).

Efficacy beliefs (self-efficacy, response efficacy, collective efficacy) were assessed with one item each. The definition and assessment of self-efficacy and collective efficacy are consistent with previous research. However, the definition of response efficacy (“response efficacy refers to the belief that one’s actions to reduce a threat will be effective” p. 7) does not match with its assessment (“My actions will influence others to behave in ways that mitigate the effects of climate change” p. 9).

For the assessment of protest enjoyment, participants were asked to respond to questions such as “how much they generally enjoy taking to the streets to protest” (p. 10). How could those participants who never participated in protest answer these questions?

In the result section, analyses are presented that are relevant to the question how Fridays for Future protesters differ from non-protesters. However, how these analyses are related to the theoretical framework is not clear.

I think it would be possible to get more out of the data. The study covers a range of interesting self-reported behaviours (protesting, meat consumption, flying) as well as a range of interesting predictors (different types of efficacy beliefs, trust, concern, enjoyment). One solution could be to follow up on the work by Whitmarsh and O’Neill (2010) who investigated whether different environmental behaviours are predicted by different variables. For instance, as protesting is a collective action, collective efficacy beliefs may predict protesting better than it predicts meat consumption and flying. If this is not the case this would be relevant as well because most models of collective action implicitly assume that collective efficacy beliefs are specifically relevant for collective action.

Hamann, K. R., & Reese, G. (2020). My influence on the world (of others): Goal efficacy beliefs and efficacy affect predict private, public, and activist pro‐environmental behavior. Journal of Social Issues, 76(1), 35-53.

Landmann, H. & Rohmann, A. (2020). Being moved by protest: Collective efficacy beliefs and injustice appraisals enhance collective action intentions for forest protection via positive and negative emotions. Journal of Environmental Psychology, https://doi.org/10.1016/j.jenvp.2020.101491

Tausch, N., & Becker, J. C. (2013). Emotional reactions to success and failure of collective action as predictors of future action intentions: A longitudinal investigation in the context of student protests in Germany. British Journal of Social Psychology, 52(3), 525–542. https://doi.org/10.1111/j.2044-8309.2012.02109.x

Whitmarsh, L., & O’Neill, S. (2010). Green identity, green living? The role of pro-environmental self-identity in determining consistency across diverse pro-environmental behaviours. Journal of Environmental Psychology, 30, 305–314. http://dx.doi.org/10.1016/j.jenvp.2010.01.003

6. PLOS authors have the option to publish the peer review history of their article (what does this mean?). If published, this will include your full peer review and any attached files.

Reviewer #1: No

Reviewer #2: No

Reviewer #3: No

---

## [Author Response · Author response to Decision Letter 0]

2 Jun 2021

Journal Requirements:

Our document is now in line with the PLOS ONE style templates. 

2) Please change "female” or "male" to "woman” or "man" as appropriate, when used as a noun (see for instance https://apastyle.apa.org/style-grammar-guidelines/bias-free-language/gender).

This has been changed. 

3) Please include additional information regarding the survey or questionnaire used in the study and ensure that you have provided sufficient details that others could replicate the analyses. For instance, if you developed a questionnaire as part of this study and it is not under a copyright more restrictive than CC-BY, please include a copy, in both the original language and English, as Supporting Information.

We have now added the German version of the questionnaire to the supplementary material.

4) Please provide additional details regarding participant consent. In the ethics statement in the Methods and online submission information, please ensure that you have specified (1) whether consent was informed and (2) what type you obtained (for instance, written or verbal, and if verbal, how it was documented and witnessed). If your study included minors, state whether you obtained consent from parents or guardians. If the need for consent was waived by the ethics committee, please include this information.

In the methods section, we now note that written consent was obtained from all participants. 

5) Please include a copy of Table 3 which you refer to in your text on page 14.

We actually meant to refer to Table 2 and have now changed this accordingly. 

6) Please include captions for your Supporting Information files at the end of your manuscript, and update any in-text citations to match accordingly. Please see our Supporting Information guidelines for more information: http://journals.plos.org/plosone/s/supporting-information.

We have now added captions for our Supporting Information file at the end of the manuscript.

Reviewer's Responses to Questions

Comments to the Author

1. Is the manuscript technically sound, and do the data support the conclusions?

Reviewer #1: Yes

Reviewer #2: Partly

Reviewer #3: Yes

2. Has the statistical analysis been performed appropriately and rigorously?

Reviewer #1: Yes

Reviewer #2: No

Reviewer #3: Yes

3. Have the authors made all data underlying the findings in their manuscript fully available?

Reviewer #1: Yes

Reviewer #2: Yes

Reviewer #3: Yes

4. Is the manuscript presented in an intelligible fashion and written in standard English?

Reviewer #1: Yes

Reviewer #2: Yes

Reviewer #3: Yes

5. Review Comments to the Author

Reviewer #1: One aim of the paper is to investigate the motives of past and intended future participation in the Friday for Future climate strikes among a sample (N = 638) of Swiss university students. The authors report that response efficacy, protest enjoyment, the perceived success of the strikes, trust in climate scientists, low trust in governments to act on climate change, female gender, and left-wing political orientation predicted strike participation, while only protest enjoyment, perceived success of the strikes and political orientation predicted intentions to join future strikes. The study’s second aim is to research the question whether there is a discrepancy between students’ participation in such collective actions like strikes and their own personal behavior. The study findings counter such accusations: Compared with the non striking students, the striking students declare eating less meat per week, having flown less in the past two years, and having sought to offset the CO2 emissions from their flights, however, there are no difference in long-haul flights. In the discussion section the authors critically discuss methodological problems of their research design and draw conclusion for future research.

Without question the paper targets an issue of great interest for the general public as well as social scientists. The paper is written in a good structured, readable way. The results concerning the personal behavior of the strike participants are convincing, however, are based on self-reported behavior, which may be biased to participants’ desire to be consistent with the behavior made salient by the survey (pro-environmental behavior).

We agree with the reviewer that self-reported measures may reflect social desirability biases. We clearly acknowledge this as a limitation in our paper and recommend that future studies measure objective behaviour. 

For me the main weakness of the paper is the not convincing theoretical part: In general the authors seems to follow a rational choice approach (Opp, Finkel), however, they also frequently refer to the social identity approach of collective behavior (e.g. Van Zomeren). The authors do not reflect the relation between these rather different theories: Whereas rational choice models stress the role of personal benefits as central determinants of collective action and stress the social dilemma nature of collective action participation (discrepancy between self- and collective interests), the social identity do not see such a dilemma because the activation of a social identity automatically motivates the person to act as a group member. Without systematically reflecting and integrating the assumptions underlying the two different theoretical frameworks, for me it seems rather eclectic to draw single constructs from different theoretical frameworks without clarifying their theoretical relation. From my point of view such a positivistic approach is not helpful for systematical theory development within the field of collective action.

We thank the reviewer for pointing this out, and we agree. As has also been noted by the other two reviewers, our theoretical justification was not convincing in the earlier draft. In response, we made major changes to the introduction. Rather than referring to specific models, we now refer to different studies that found the assessed variables to be of significance for collective action. Since we now do not base our theoretical background on these two different theoretical models anymore, this eliminated further explanations on the differences between the models. Further, we included more recent studies on the climate strikes in our introduction which were not yet published at the time we submitted our manuscript. This also helped to enrich our theoretical part. We believe that these changes significantly improved the quality of our manuscript and made our theoretical part more convincing. We hope that the reviewer agrees with these changes.

On the empirical level the used construct measure reflect the paper’s weak theoretical reflection. For example the item “enjoy being part of a movement” could also be seen as a typical indicator of the social identity construct. In this case the empirical results would indicate that social identity is the main predictor of collective action participation. A result replicating the findings of dozens of studies conducted within the social identity framework (see e.g. the work of van Zomeren). Also the discussion of the concepts of self- and response efficacy is not in line with the current theoretical discussion, I would recommend the authors to read the work of van Zomeren on the new concept of participatory efficacy (e.g., Van Zomeren, M., Saguy, T., & Schellhaas, F. M. (2013). Believing in “making a difference” to collective efforts: Participative efficacy beliefs as a unique predictor of collective action. Group Processes & Intergroup Relations, 16(5), 618-634).

We thank the reviewer for this comment. We see the reviewer’s point that the item “being part of a movement” could also be seen as reflecting social identity perceptions. We now mention in our discussion section that this item could also be interpreted as measuring social identity, and that the fact that we did not assess social identity perceptions presents a significant limitation of our work. We also thank the reviewer for pointing out the literature on participatory efficacy. Given these comments and the comments by the other two reviewers, we updated the theoretical discussion of our efficacy items and recommend that future research on the climate strikes assess participatory efficacy.

Reviewer #2: This manuscript addresses a timely, interesting and still under-researched topic. The purpose and significance of the study are clearly stated in the introduction section. The authors provide sufficient information to understand the study goals and the approach followed, and a sufficient and detailed discussion is provided. Despite its potential, I believe this manuscript, at least in its current form, is not ready for publication. My major issues with it are related to the theoretical section and with some methodological limitations. I explain these issues below:

1. While the manuscript addresses some relevant literature, there is an overall trend towards selectiveness and to superficially discuss theoretical models and approaches. I believe this is misleading and a simplification in at least two ways. First, while there is some empirical evidence suggesting that young people – especially when compared to previous generations (see Maria Teresa Grasso’s work) – are more disengaged from conventional and traditional forms of political participation (e.g., voting) there is no consensus in the literature regarding other forms of participation. The myth of youth apathy has been amply discussed in the literature and it has evolved towards a more comprehensive view of the forms, patterns and levels of young people’s participation. The way previous literature is presented in this manuscript suggests a consensual view of young people as apathetic and politically disengaged, but that does not correspond to the current debates in the literature. Secondly, the manuscript also suggests a view of young people's political participation as a matter of being politically active or inactive. Such a dichotomy is again a simplification considering that previous studies have shown that even a passive stance can actually mask a latent political involvement. I strongly recommend further engagement with current debates on the field of youth participation. See for example:

- Amnå, E., & Ekman, J. (2014). Standby citizens: Diverse faces of political passivity. European Political Science Review,6(2), 261–281. https://doi.org/10.1017/S175577391300009X

- Cammaerts, B., Bruter, M., Banaji, S., Harrison, S., & Anstead, N. (2014). The myth of youth apathy: Young Europeans’ critical attitudes toward democratic life. American Behavioral Scientist, 58(5), 645–664. https://doi.org/10.1177/0002764213515992

- Barret, M. & B. Zani, B. (Eds.). Political and civic engagement: multidisciplinary perspectives (pp. pp. 499-512). Surrey: Routledge.

- Weiss, J. (2020). What Is Youth Political Participation? Literature Review on Youth Political Participation and Political Attitudes. Front. Polit. Sci., 2020. https://doi.org/10.3389/fpos.2020.00001

- Earl, J., Maher, T. V., and Elliott, T. (2017). Youth, activism, and social movements. Sociol. Compass 11, e12465. https://doi.org/10.1111/soc4.12465

2. Despite the extensive amount of literature on the topic of young people’s participation, there is still a need for studies focusing on the motives for collective action, and participation in climate strikes do mark a shift in political engagement by young people, as argued by the authors. However, this needs to be framed more carefully, considering past and current debates within the field.

We thank the reviewer for comments 1 and 2 pointing out limitations in our reporting of youth apathy. We also thank the reviewer for suggesting these articles on youth participation, which we found very helpful. We have adapted this section in the introduction accordingly and believe it improved the whole introduction. Specifically, we now mention that the youth might not necessarily have lower levels of interest in political topics than the rest of the population, but that they are instead using different practices of political engagement (e.g., boycotts and buycotts) compared to mainstream practices. 

3. Still related to the literature review, it was rather surprising to see that the theoretical concepts (and model) used to design the study and the hypotheses were based on the literature on collective action. Again, there are plenty of studies on the motivations for young people’s participation that could help to design and understand the findings. In addition, there is also some confusion when discussing the collective interest model, the socio-psychological collective action model, and the dual path model (in the discussion). These models, if relevant, should be better presented.

We agree with the reviewer, as well as with the other two reviewers, who all mentioned that the theoretical background and the presentation of the different models were not presented very well in the first draft. We thus made significant changes to the introduction to make the theoretical background clearer. Rather than referring to specific models, we now refer to different studies that found the assessed variables to be of significance for collective action. We also added studies on the motivations of young people’s participation, specifically also studies on the Fridays for Future strikes (see reply to next comment).

4. The authors argued that there is only one peer-reviewed study that has been conducted on the Fridays for Future strikes (line 135). I wonder why the following literature was not considered (see below) as they all were peer-reviewed and published in 2020:

- Mari Martiskainen, Stephen Axon, Benjamin K. Sovacool, Siddharth Sareen, Dylan Furszyfer Del Rio, Kayleigh Axon (2020). Contextualizing climate justice activism: Knowledge, emotions, motivations, and actions among climate strikers in six cities. Global Environmental Change. https://doi.org/10.1016/j.gloenvcha.2020.102180.

- Emilsson K, Johansson H, Wennerhag M. Frame Disputes or Frame Consensus? (2020). “Environment” or “Welfare” First Amongst Climate Strike Protesters. Sustainability. 2020; 12(3):882. https://doi.org/10.3390/su12030882

- Brügger A, Gubler M, Steentjes K, Capstick SB. (2020). Social Identity and Risk Perception Explain Participation in the Swiss Youth Climate Strikes. Sustainability. 2020; 12(24):10605. https://doi.org/10.3390/su122410605

We thank the reviewer for pointing us to these highly relevant publications. Two of these were published after we submitted our article in October 2020, and we missed the third one. We now refer to these, as well as to other papers on the strikes.

5. There is a need for a better articulation between the empirical part and the theoretical approach. The authors should provide more details for their statistical and methodological decisions. Similarly, some of the expected results need better contextualization. For example, little is said about the role of trust and protest enjoyment in the theoretical section. Both dimensions are however key in this study. Specifically, perceived enjoyment of protest and related constructs are almost absent from the literature review section, but it is then mentioned in the discussion. I suggest introducing all the theoretical constructs in the theoretical section.

We thank the reviewer for pointing out the fact that some results need better contextualization and that not all theoretical constructs were present in the theoretical section. We added more information on the role of trust in climate scientists and governments and protest enjoyment in the theoretical section and linked our results to these previous studies in the discussion section. 

6. In my opinion, there are several strong limitations with the measures used. First, most dimensions are composed of one item (e.g., trust in climate scientists, self-efficacy, response efficacy, collective efficacy). Second, the authors created their own scales (e.g., climate change concern, protest enjoyment, pro-environmental behaviour) and there is only one reference for an item (self-efficacy item). I believe this needs to be justified, because there are already validated scales for the constructs measured. Third, while developing your own scales could be acceptable, in that case more rigorous statistical procedures should have been followed. In this regard, I would recommend to (at least) present the results of the exploratory factor analysis scales such as “climate concern scale” and “protests enjoyment”. As far as I understood, both were developed by the authors. I also wonder why pro-environmental behaviours were not considered as a composite variable for the analysis.

We agree with the review that more rigorous statistical procedures are in order given that we developed some scales. For the climate change concern scale, we verified sampling accuracy with the Kaiser-Meyer-Olkin (KMO) measure which resulted in a value of 0.8 (above the commonly recommended value of 0.60). Bartlett's test of sphericity further indicated sufficient significant correlation in the data for factor analysis (χ2 (15) = 1014.74, p < .001). A screeplot indicated that only one component had an eigenvalue larger than 1, supporting our approach. Further, all items had factor loadings > 0.60 except for worry about extreme weather events (0.45) and human migration (0.49). The scale also had an acceptable Cronbach alpha of 0.78. 

Regarding our protest enjoyment scale, we verified sampling accuracy with the Kaiser-Meyer-Olkin (KMO) measure which resulted in a value of 0.6. Bartlett's test of sphericity further indicated sufficient significant correlation in the data for factor analysis (χ2 (3) = 421.08, p < .001). A factor analysis showed that the first two items had adequate factor loadings (Item 1 = 0.76, Item 2 = 0.84), whereas Item 3 resulted in a factor loading of 0.41. However, given the conceptual similarity of the items and an acceptable Cronbach alpha = 0.70, the use of this scale is warranted. 

The reason for not using a composite scale of pro-environmental behaviours is that meat consumption and flying were assessed on a scale from 0 = never to 6 = more than five times), while we asked participants whether they had offset flight emissions and donated money with a yes/now response option. 

7. There is also an overall lack of rigour when presenting the “collective efficacy” construct and also with the way efficacy is measured. While the authors recognise the limitations of their variable “collective efficacy”, there are other aspects that should be considered. For example, group efficacy, perceived efficacy of collective action and collective efficacy are different concepts and should not be used as if they mean the same thing. In addition, considering that what was measured in collective efficacy is not even related to collective action, I would suggest avoiding claiming that “the result that collective efficacy was not a predictor of participation was surprising” or that “it doesn't fit the literature”. Probably, this finding may only represent a problem with the way the variable was measured.

We agree with the reviewer that this finding likely relates to how we assessed collective efficacy. We previously acknowledged this with a sentence in our discussion section but have now put more emphasis on this. On pages 20-21, we present different explanations for why our item on collective efficacy was not a significant predictor. Specifically, we also refer to the recent study on the Fridays for Future climate strikes by Wallis and Loy (2021) who also found collective efficacy not to be a significant predictor of strike participation.

We then specifically state: “For all these reasons, we discourage future use of our item to assess collective efficacy and advise future research to assess collective efficacy specifically related to the climate strikes and not just to climate change in general.” 

Wallis, H., & Loy, L. S. (2021). What drives pro-environmental activism of young people? A survey study on the Fridays For Future movement. Journal of Environmental Psychology, 74, 101581. https://doi.org/10.1016/j.jenvp.2021.101581

8. I would recommend being more cautious in the interpretation of findings considering all the mentioned methodological limitations of this study. The contributions of the study should be more clearly articulated with the findings, and in the discussion section the authors should try to avoid just merely repeating the findings. Besides, the interpretations and suggestions made in the discussion (e.g., for example regarding the role and the multiple facets of efficacy in collective action) should be better supported by previous literature. I would also be careful in providing recommendations for future studies and claiming that there is none research on the topic. For example, there are some studies on the role of positive emotions and sense of obligation in collective action that could be used to understand the findings. As such, the recommendations for future studies should at least recognise that these already exist.

We thank the reviewer for this comment. We agree that some arguments in our discussion necessitated of better support by the literature. We have considerably rewritten our discussion section and referred to more literature to better support our discussion points. Specifically, we now refer to literature on the role of positive emotions (in the introduction as well as the discussion section) and have expanded our discussion on the role of different facets of efficacy. We hope that the reviewer will find these changes satisfying. 

9. Implications of the research – in the introduction the authors proposed to extend the literature on collective action. This should be more clearly addressed in the final section of the manuscript.

We also agree with the reviewer on this point and added the following paragraph at the end of the discussion section: 

“Our study thus advances the literature on collective action in several ways. First, we show that trust in climate scientists is an important determinant for participation in the climate strikes. While the importance of trust in science and scientists for youth engagement has been acknowledged in the literature (63), this is the first study to show its importance for climate strike participation. Second, we show that low trust in governments (both national and international) to act on climate change positively predicts youth participation in climate strikes. As previous studies found trust in political institutions to positively predict individual pro-environmental behaviours and policy support (45,46), the opposite seems to be true for climate activism. We recommend future studies to consider the role of trust in different actors for protest participation. Third, we provide further evidence that the role of positive emotions is a strong predictor of protest participation, and in our case, an even stronger predictor than negative emotions such as worry about climate change. Fourth, we show that predictors differ between participation and future intended participation, thus shedding light on the intention-behaviour gap regarding strike participation.”

10. Some minor issues: 1) See line 143 – “As described by Finkel and Müller 10/20/2020 8:34:00 AM”; 2) Why was prosocial behaviour considered a keyword?

Thank you for pointing out these minor issues. We have now deleted the word "prosocial" as a keyword and fixed the reference issue on line 143.

That said, we wanted to thank the reviewer for their many relevant comments. The inclusion of all their recommendations and suggestions led to significant changes throughout the manuscript, especially to the re-writing of much of the introduction and discussion section, and we believe these changes to have significantly improved the quality of this article. 

Reviewer #3: In the manuscript entitled “To strike or not to strike? An investigation of the determinants of strike participation at the Fridays for Future climate strikes in Switzerland” a study with Fridays for Future protesters and non-protesters is reported. The manuscript taps at a timely topic. However, due to the following points I cannot recommend to publish the manuscript in its current form.

The theoretical framework should be clarified. For instance, the authors refer to the social identity model of collective action and derive predictions from that model (p. 5). Then they refer to the collective interest model (p. 6 f.) and derive predictions from that model. However, it is not sufficiently clear how these models and predictions are related. The authors also report that they assess trust in scientists (p. 7) and that previous research suggests that pleasure individuals get from protesting also predicts participation (p. 8) but they do not explain how they expect these variables/processes to be linked with the proposed model.

We are grateful that all reviewers mentioned that the theoretical framework needs to be clarified. We agree that the previous version caused for some confusion. We thus made significant changes to the introduction to make the theoretical background clearer. Rather than referring to specific models, we now refer to different studies that found the assessed variables to be of significance for collective action. Further, we included more recent studies on the climate strikes in our introduction which were not yet published at the time we submitted our manuscript. We believe that this article significantly benefitted from the major revision of our introduction and hope that reviewers agree.

The authors introduce the distinction between emotion-focused coping and problem-focused coping, which was proposed for the context of collective action by Van Zomeren and colleagues (p. 5). However, a recent study shows that the so-called problem-focused path to collective action is mediated via emotions as well (Landmann & Rohmann, 2020). The authors may want to update their literature review including the role of positive emotions for collective action (Tausch & Becker, 2013) as well as different types of efficacy beliefs for environmental activism (Hamann & Reese, 2020).

We thank the reviewer for pointing out this very relevant literature. On page 6, we now point out that the problem-focused coping approach has been challenged by recent findings by Landmann & Rohmann (2020). We also included these and other papers about the role of positive emotions for collective action. We also found the paper by Hamann & Reese (2020) very useful, which also helped to better define and contextualize our different types of efficacy. 

Efficacy beliefs (self-efficacy, response efficacy, collective efficacy) were assessed with one item each. The definition and assessment of self-efficacy and collective efficacy are consistent with previous research. However, the definition of response efficacy (“response efficacy refers to the belief that one’s actions to reduce a threat will be effective” p. 7) does not match with its assessment (“My actions will influence others to behave in ways that mitigate the effects of climate change” p. 9).

We thank the reviewer for pointing this out and agree that our previous definition did not accurately reflect our assessed item. The reason we formulated this item this way, was that we believed that protesters would be more likely to participate at the climate strikes if they believed that their behaviour (including strike participation) had the potential to influence others to behave in ways that mitigate the effects of climate change. More precisely, we assessed if participants expected that others would also behave environmentally friendly as a response to their own actions, e.g., strike participation. We now see that it would have been better to specifically refer to the climate strikes in all our efficacy items, rather than just to climate change in general. We explicitly mention this as limitation of our study. We now slightly adjusted our definition of response efficacy (“response efficacy refers to the belief that one’s actions to achieve a certain goal will be effective (25)”) and now refer to the work by Hamann & Reese (2020) on indirect goal efficacy, which is what we measured here. We hope that the section on response efficacy is clearer now and wanted to thank the reviewer again for pointing us to this very useful literature. Following the advice provided by reviewer 1, we now also mention that future studies should additionally measure participatory efficacy, as van Zomeren et al., (2013) found participatory efficacy to be distinct type of efficacy that represents a unique predictor of collective action.

Van Zomeren, M., Saguy, T., & Schellhaas, F. M. (2013). Believing in “making a difference” to collective efforts: Participative efficacy beliefs as a unique predictor of collective action. Group Processes & Intergroup Relations, 16(5), 618-634). 10.1177/1368430212467476

For the assessment of protest enjoyment, participants were asked to respond to questions such as “how much they generally enjoy taking to the streets to protest” (p. 10). How could those participants who never participated in protest answer these questions?

We see the reviewer’s point. We find it likely that participants who have never done that behavior would answer this in the hypothetical, that is, whether they would enjoy taking to the streets to protest. Therefore, we agree this wording isn't ideal, but we suggest the item is still capturing something useful and consistent for all participants.

In the result section, analyses are presented that are relevant to the question how Fridays for Future protesters differ from non-protesters. However, how these analyses are related to the theoretical framework is not clear.

I think it would be possible to get more out of the data. The study covers a range of interesting self-reported behaviours (protesting, meat consumption, flying) as well as a range of interesting predictors (different types of efficacy beliefs, trust, concern, enjoyment). One solution could be to follow up on the work by Whitmarsh and O’Neill (2010) who investigated whether different environmental behaviours are predicted by different variables. For instance, as protesting is a collective action, collective efficacy beliefs may predict protesting better than it predicts meat consumption and flying. If this is not the case this would be relevant as well because most models of collective action implicitly assume that collective efficacy beliefs are specifically relevant for collective action.

We thank the reviewer for these helpful suggestions. We would like to gently decline the invitation to expand the paper in these ways for a couple reasons. First, these behaviors are not the main focus. Second, those behavior measures are not representative of all pro-environmental behaviors or a coherent, a priori-designed subset. Third, their measurement operationalization isn't the most robust and therefore we suggest not placing them more centrally in the narrative. We totally agree this sort of paper would be valuable, but it doesn't fit great within the current scope.

Hamann, K. R., & Reese, G. (2020). My influence on the world (of others): Goal efficacy beliefs and efficacy affect predict private, public, and activist pro‐environmental behavior. Journal of Social Issues, 76(1), 35-53.

Landmann, H. & Rohmann, A. (2020). Being moved by protest: Collective efficacy beliefs and injustice appraisals enhance collective action intentions for forest protection via positive and negative emotions. Journal of Environmental Psychology, https://doi.org/10.1016/j.jenvp.2020.101491

Tausch, N., & Becker, J. C. (2013). Emotional reactions to success and failure of collective action as predictors of future action intentions: A longitudinal investigation in the context of student protests in Germany. British Journal of Social Psychology, 52(3), 525–542. https://doi.org/10.1111/j.2044-8309.2012.02109.x

Whitmarsh, L., & O’Neill, S. (2010). Green identity, green living? The role of pro-environmental self-identity in determining consistency across diverse pro-environmental behaviours. Journal of Environmental Psychology, 30, 305–314. http://dx.doi.org/10.1016/j.jenvp.2010.01.003

---

## [Decision Letter · Decision Letter 1]

31 Aug 2021

To strike or not to strike? An investigation of the determinants of strike participation at the Fridays for Future climate strikes in Switzerland

PONE-D-20-32959R1

Dear Dr. Cologna,

We’re pleased to inform you that your manuscript has been judged scientifically suitable for publication and will be formally accepted for publication once it meets all outstanding technical requirements. I would like to apologise for the delay in reaching this decision, I hope you can appreciate it is not always easy to obtain reviews from experts in the field, given the voluntary nature of peer review.

Kind regards,

Jim Uttley, Ph.D

Academic Editor

PLOS ONE

Additional Editor Comments (optional):

Reviewers' comments:

Reviewer's Responses to Questions

**Comments to the Author**

1. If the authors have adequately addressed your comments raised in a previous round of review and you feel that this manuscript is now acceptable for publication, you may indicate that here to bypass the “Comments to the Author” section, enter your conflict of interest statement in the “Confidential to Editor” section, and submit your "Accept" recommendation.

Reviewer #2: All comments have been addressed

2. Is the manuscript technically sound, and do the data support the conclusions?

Reviewer #2: Yes

3. Has the statistical analysis been performed appropriately and rigorously? 

Reviewer #2: I Don't Know

4. Have the authors made all data underlying the findings in their manuscript fully available?

Reviewer #2: Yes

5. Is the manuscript presented in an intelligible fashion and written in standard English?

Reviewer #2: Yes

6. Review Comments to the Author

Reviewer #2: Overall, I believed the manuscript was much improved, particularly in the theoretical framework section. There are still some critical limitations in this study, as highlighted in the first review, but these limitations cannot be properly addressed.

7. PLOS authors have the option to publish the peer review history of their article (what does this mean?). If published, this will include your full peer review and any attached files.

Reviewer #2: No

---

## [Editor Report · Acceptance letter]

7 Oct 2021

PONE-D-20-32959R1 

To strike or not to strike? An investigation of the determinants of strike participation at the Fridays for Future climate strikes in Switzerland 

Dear Dr. Cologna:

I'm pleased to inform you that your manuscript has been deemed suitable for publication in PLOS ONE. Congratulations! Your manuscript is now with our production department. 

Kind regards, 

on behalf of

Dr. Jim Uttley 

Academic Editor

PLOS ONE